# Kinetic resolution of indolines by asymmetric hydroxylamine formation

Gang Wang[1], Ran Lu[2], Chuangchuang He[2] & Lei Liu [1,2]✉

Catalytic kinetic resolution of amines represents a longstanding challenge in chemical synthesis. Here, we described a kinetic resolution of secondary amines through oxygenation to produce enantiopure hydroxylamines involving N–O bond formation. The economic and practical titanium-catalyzed asymmetric oxygenation with environmentally benign hydrogen peroxide as oxidant is applicable to a range of racemic indolines with multiple stereocenters and diverse substituent patterns in high efficiency with efficient chemoselectivity and enantio-discrimination. Late-stage asymmetric oxygenation of bioactive molecules that are otherwise difficult to synthesize was also explored.

[1] School of Chemistry and Chemical Engineering, Shandong University, Jinan, China. [2] School of Pharmaceutical Sciences, Jinan, China. ✉email: leiliu@sdu.edu.cn

Enantiopure cyclic secondary amines are key constituents of natural products, pharmaceuticals, and agricultural chemicals[1]. Catalytic kinetic resolution (KR) of racemic amines represents a practical and robust approach to access optically pure targets, especially in cases where the racemates are readily available and the enantiopure materials are not[2–8], though the theoretical yield of the expected optically pure target can never exceed a limit of 50%[9–11]. The current nonenzymatic KR of secondary amines predominantly relies on asymmetric N-acylation strategy involving N–C bond formation, which typically requires the use of stoichiometric pre-prepared acylating agents involving lengthy reagent synthesis (Fig. 1a)[12–18]. Development of a catalytic KR of secondary amines based on other elementary reaction bearing an economy-oriented mind-set would be highly desirable.

Oxygen atom transfer (OAT) reaction is ubiquitous in biological systems, organic synthesis, and industrial processes[19–23]. Current asymmetric OAT studies predominantly focused on oxygenation of alkenes and sulfides involving C–O and S–O bond formation[24–34]. We envisioned that asymmetric OAT to secondary amines to produce enantiopure hydroxylamines involving N–O bond formation would be an ideal template for KR design based on economical and environmental factors (Fig. 1b). However, asymmetric OAT to amines has remained a formidable challenge[35,36]. To the best of our knowledge, asymmetric oxygenation of secondary amines to produce hydroxylamines through either enzymatic or non-enzymatic catalysis has never been established to date, which might be ascribed to three key challenges. First, the inherent high reactivity of amines results in easy nonselective oxygenation without the intervention by a catalyst. Second, competitive dehydrogenation of secondary amines to imines usually accompanies the expected OAT process[37–39]. Third, chirality in hydroxylamine products could be facilely destroyed through further oxidation to nitrones[40]. On the other hand, aqueous hydrogen peroxide is a desirable oxidant from the viewpoints of atom efficiency (48%), easy-to-handle, and ecological benignity[32,41]. Given the significance of optically pure indolines in modern pharmacology, we herein report a titanium-catalyzed KR of indolines using $H_2O_2$ as the oxidant through N–O bond formation. Late-stage asymmetric oxygenation of bioactive molecules that are otherwise difficult to synthesize was also explored.

## Results

**Reaction condition optimization.** Initially, asymmetric oxygenation of racemic indoline **1a** was selected as a reference reaction using aqueous $H_2O_2$ as the oxo-transfer agent to search for a suitable chiral catalyst (Table 1). Chiral monomeric (salen)titanium(IV) **C1** exhibited no oxidation catalysis reactivity (entry 1). We then explored dimeric metallosalen complex as catalyst. Delightedly, di-μ-oxo titanium(salen) **C2** effected the expected asymmetric oxygenation, though poor chiral recognition and notable over-oxidation were observed (entry 2). The substituent patterns on 1,2-ethanediamine proved to be crucial to catalytic reactivity and asymmetric induction. Replacing the 1,2-cyclohexanediamine moiety in **C2** with 1,2-diphenylethylenediamine one (**C3**) significantly reduced the oxidation catalytic reactivity (entry 3). Di-μ-oxo titanium(salalen) **C4** was prepared by reducing one of the process (entry 4). Displacing the phenyl group on salicylaldehydes with other two imine bonds of **C2**, was beneficial for suppressing undesired over-oxidation substituents afforded inferior results, which prompted us to introduce another chiral element at $C_3$ site of the basal salalen ligand to enhance the enantio-differentiating ability of the catalyst. The "hybrid" titanium(salalen) **C5** bearing a ($R_a$)-binaphthyl unit on the imine side was not an effective catalyst (entry 5). Promising chiral recognition was observed when di-μ-oxo titanium(salalen) catalyst **C6** bearing two ($R_a$)-binaphthyl units on both amine and imine sides were used (entry 6). The absolute configuration of (R)-**1a** was assigned to be R by comparing the optical rotation and HPLC analysis with reported data. See the Supplementary Information for details. Reversing the absolute configuration of 1,2-cyclohexanediamine in catalyst **C7** provided a mismatch recognition (entry 7). Extensive optimization of the solvent identified $CHCl_3$ to be optimal (entries 8–10). The level of chiral discrimination was further enhanced by lowering the loading of titanium(salalen) **C6** from 2.5 mol % to 1.0 mol %, though a slightly prolonged time period was required (entry 11).

a) Asymmetric acylation strategy via N-C bond formation (*Fu, Hou, Bode, Kozlowski*)

b) Asymmetric oxygenation strategy via N-O bond formation (*this work*)

***challenges:***

nonselective background reaction

competitive chemoselectivity

readily over-oxidation

**Fig. 1 Overview of KR of cyclic secondary amines. a** Asymmetric acylation strategy via N–C bond formation. **b** Asymmetric oxygenation strategy via N–O bond formation.

**Table 1 Reaction condition optimization[a].**

| Entry | Cat. | t (h) | Yield (%)[b] 1a/2a/3 | ee (%)[c] (R)-1a/(S)-2a | s[d] |
|---|---|---|---|---|---|
| 1 | C1 | 24 | > 95/< 5/< 5 | n.d. | n.d. |
| 2 | C2 | 12 | 44/23/27 | 35/30 | 2.5 |
| 3 | C3 | 24 | > 95/< 5/< 5 | n.a. | n.a. |
| 4 | C4 | 1 | 48/32/13 | 43/45 | 3.9 |
| 5 | C5 | 24 | 58/16/20 | 35/54 | 4.7 |
| 6 | C6 | 8 | 49/35/10 | 74/76 | 16 |
| 7 | C7 | 12 | 78/9/8 | 7/30 | 2 |
| 8[e] | C6 | 12 | 67/17/9 | 37/90 | 28 |
| 9[f] | C6 | 12 | 51/36/9 | 65/75 | 14 |
| 10[g] | C6 | 4 | 47/43/5 | 94/89 | 60 |
| 11[g,h] | C6 | 6 | 47/45/3 | 92/92 | 79 |

[a]Reaction condition: rac-**1a** (0.1 mmol), 30% aqueous $H_2O_2$ (0.1 mmol), and catalyst (2.5 mol %) in $CH_2Cl_2$ (1.0 mL) at rt for indicated time period, unless otherwise noted. [b]Yield of isolated product. [c]Determined by HPLC analysis on a chiral stationary phase. [d]Selectivity (s) values were calculated through the equation $s = \ln[(1-C)(1-ee_{1a})]/\ln[(1-C)(1+ee_{1a})]$, where C is the conversion; $C = ee_{1a}/(ee_{1a} + ee_{2a})$. [e]Ethyl acetate as solvent. [f]$CH_3CN$ as solvent. [g]$CHCl_3$ as solvent. [h]1 mol % of **C6** used.

**Asymmetric oxygenation of indolines bearing one stereocenter.** The scope of di-μ-oxo titanium(salalen)-catalyzed asymmetric oxygenation of racemic indolines was explored (Fig. 2)[40]. Substrates **1a**-**1k** bearing a wide range of electronically varied aryl and heteroaryl groups at $C_2$ position with different substituent patterns proceeded with excellent chemoselectivity and chiral recognition (Fig. 2a). Indolines **1l**-**1s** bearing diverse $C_2$-alkyl substituents were suitable components with excellent selectivity factors (Fig. 2b). Diverse functional groups including phenyl motif (**1o** and **1p**), silyl ether (**1q**), carboxylic acid ester (**1r**), and terminal alkyne (**1s**) were tolerated as additional functional handle. Spirocyclic **1r** and **1s** containing the variant of the geminal disubstitution at $C_3$-position were also tolerated (Fig. 2c). Simple indoline **1t** without $C_3$-substituent was competent substrate, though the oxidized hydroxylamine **2t** was unstable and underwent decomposition during purification probably due to the existence of reactive benzylic $C_3$–H bonds.

The substituent effects on the indoline arene were then investigated (Fig. 3). A range of electron-withdrawing and -donating substituents at either $C_4$, $C_5$, or $C_6$ position of substrates **4a**–**4j** were tolerated with high level of chiral discrimination (Fig. 3). Bromo (**4a**–**4c**), chloro (**4d**), and fluoro (**4e**) substituents were compatible with the oxidation system for further diversification.

**Scope of indolines bearing two stereocenters.** The success in asymmetric oxygenation of racemic indolines bearing one stereocenter prompted us to further investigate the tolerance of substrates bearing two stereocenters (Fig. 4)[42–47]. Enantio-differentiating oxygenation of *trans*-2,3-trisubstituted indoline rac-**6a** bearing $C_3$ quaternary chiral center proceeded smoothly, providing (2R, 3S)-**6a** in 46% yield with 88% ee together with hydroxylamine (2S, 3R)-**7a** in 45% yield with 93% ee (s = 81). *Cis*-2,3-trisubstituted indoline rac-**6b** was also compatible with the asymmetric oxygenation conditions (s = 46). Common functional groups at the $C_3$ quaternary center, like aryl (**6c**-**6f**) and cyano (**6f** and **6g**), were well tolerated with good selectivity factors of 49–87. Cyclohexane-fused indolines, key structural motifs in a number of *Aspidosperma* alkaloids, were competent

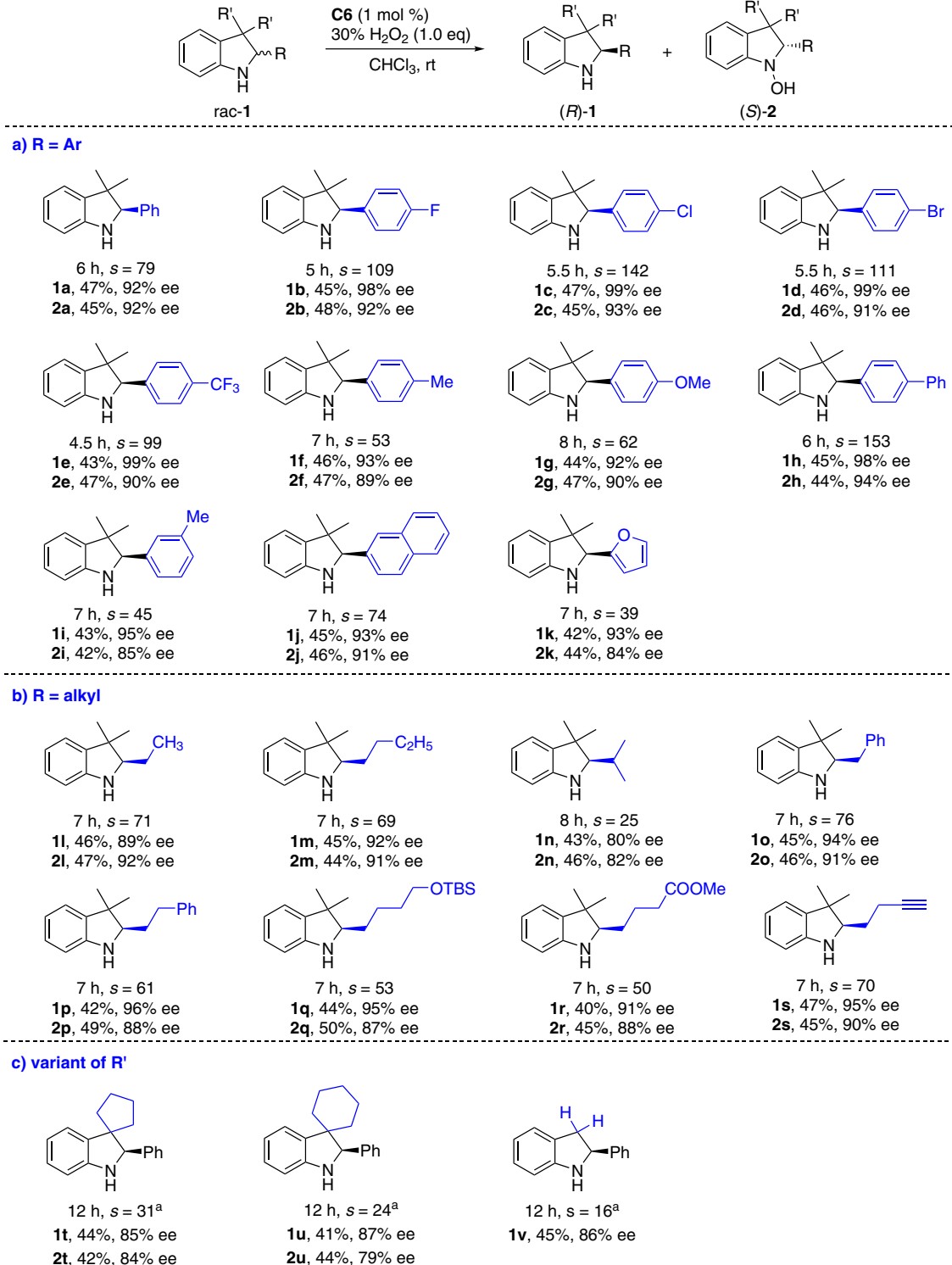

**Fig. 2 Asymmetric oxygenation of racemic C₂-substituted indolines.** Conditions: rac-**1** (0.1 mmol), 30% aqueous $H_2O_2$ (0.1 mmol), and **C6** (1 mol %) in CHCl₃ (1.0 mL) at rt for indicated time period. [a]Reaction with 2.0 equiv of aqueous $H_2O_2$ at 0 °C.

substrates, as demonstrated by effective enantio-differentiating oxygenation of rac-**6h** and **6i**[48–50]. *Trans*-2,3-disubstituted indoline rac-**6j** participated in asymmetric oxygenation, and (2S, 3S)-**6j** was recovered in 42% yield with 84% ee. Due to the existence of reactive benzylic C₃–H bond, the oxidized hydroxylamine **7j** was not stable under the oxidation conditions, and was further oxidized to several unexpected compounds (see the Supplementary Information for details)[51].

**Synthetic applications**. The synthetic utilities of the method were next examined (Fig. 5). The optically pure recovered indolines and oxidized hydroxylamines can undergo interconversion with the ee highly conserved. Hydroxylamine (S)-**2a** was reduced to (S)-**1a** in the presence of Zn and AcOH (Fig. 5a). The absolute configuration of (S)-**2a** was assigned to be S by HPLC analysis of the reduction product (S)-**1a**. See the Supplementary Information for details. Indoline (R)-**1a** was selectively oxidized to

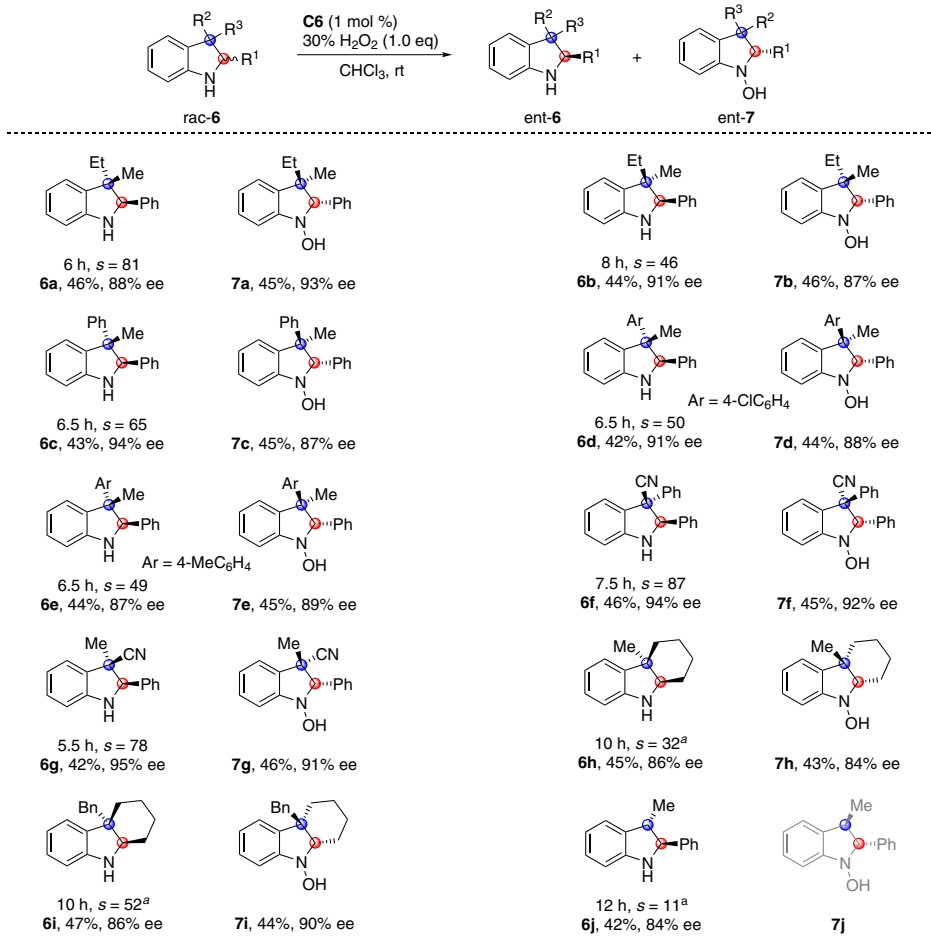

**Fig. 3 Substituent effects on the indoline arene.** Conditions: rac-**4** (0.1 mmol), 30% aqueous $H_2O_2$ (0.1 mmol), and **C6** (1 mol %) in $CHCl_3$ (1.0 mL) at rt for indicated time period.

**Fig. 4 Scope of indolines bearing two stereocenters.** Conditions: rac-**6** (0.1 mmol), 30% aqueous $H_2O_2$ (0.1 mmol), and **C6** (1 mol %) in $CHCl_3$ (1.0 mL) at rt for indicated time period. [a]Reaction with 2.0 equiv of aqueous $H_2O_2$ at 0 °C.

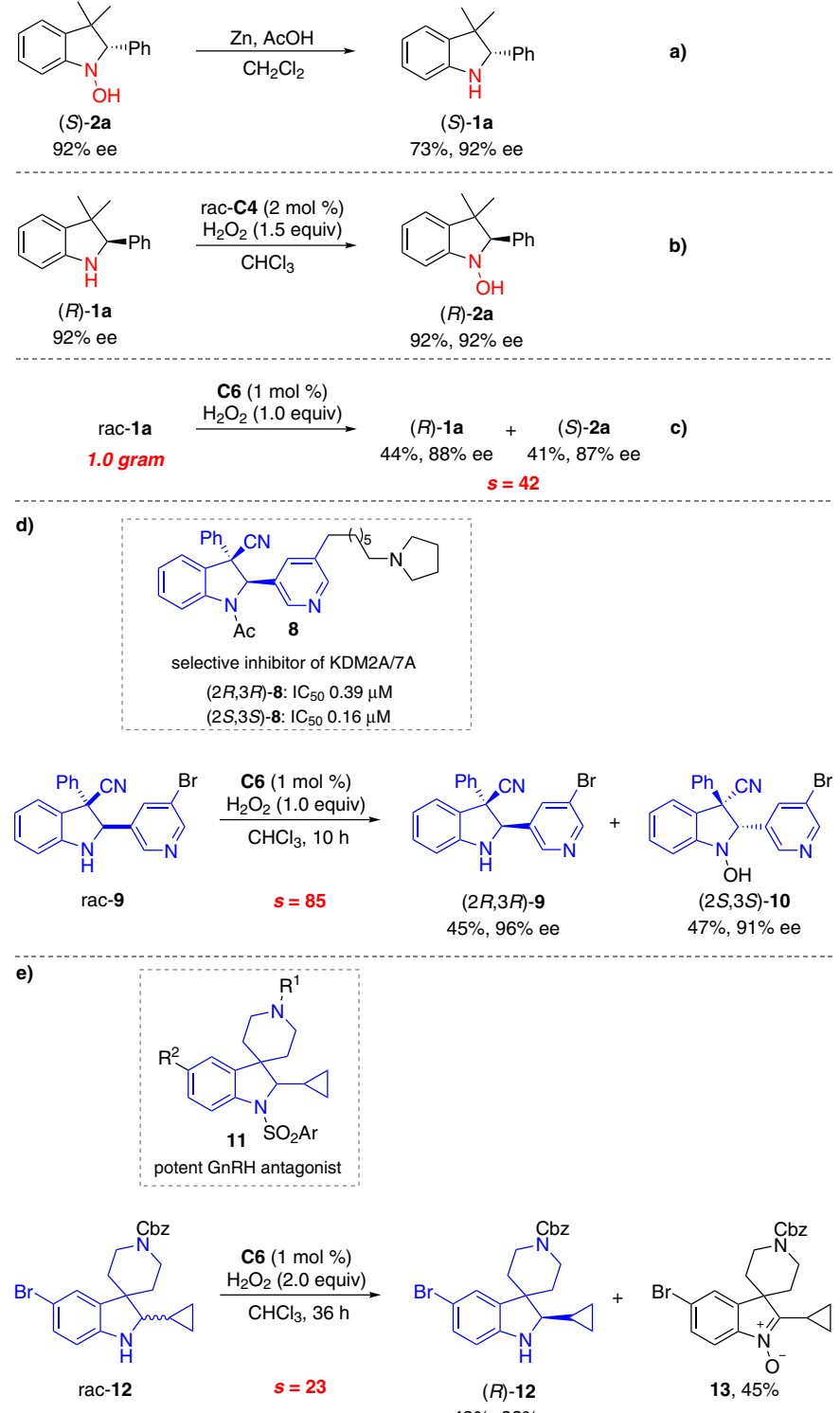

**Fig. 5 Synthetic applications. a** Reduction of hydroxylamine (*S*)-**2a**. **b** Oxidation of indoline (*R*)-**1a**. **c** Gram-scale experiment. **d** Late-stage enantio-differentiating oxygenation intermediates of KDM2A/7A. **e** Late-stage enantio-differentiating oxygenation intermediates of GnRH antagonists.

hydroxylamine (*R*)-**1a** in 92% yield when the combination of racemic titanium(salalen) **C4** and aqueous $H_2O_2$ was employed (Fig. 5b). Severe over-oxidation to nitrone was observed when common oxidants such as 3-chloroperoxybenzoic acid (mCPBA), $NaWO_4/H_2O_2$, and methyltrioxorhenium (MTO)/$H_2O_2$, were used. The reaction in a gram-scale proceeded without obvious

loss of enantioselectivity (Fig. 5c). The late-stage enantio-differentiating oxygenation advanced intermediates of bioactive molecules that would be otherwise difficult to access was further explored. *Trans*-2,3-trisubstituted indoline **8** was identified as a potent and selective inhibitor of the histone lysine demethylases KDM2A/7A (Fig. 5d)[52]. Under the standard conditions, rac-**9**

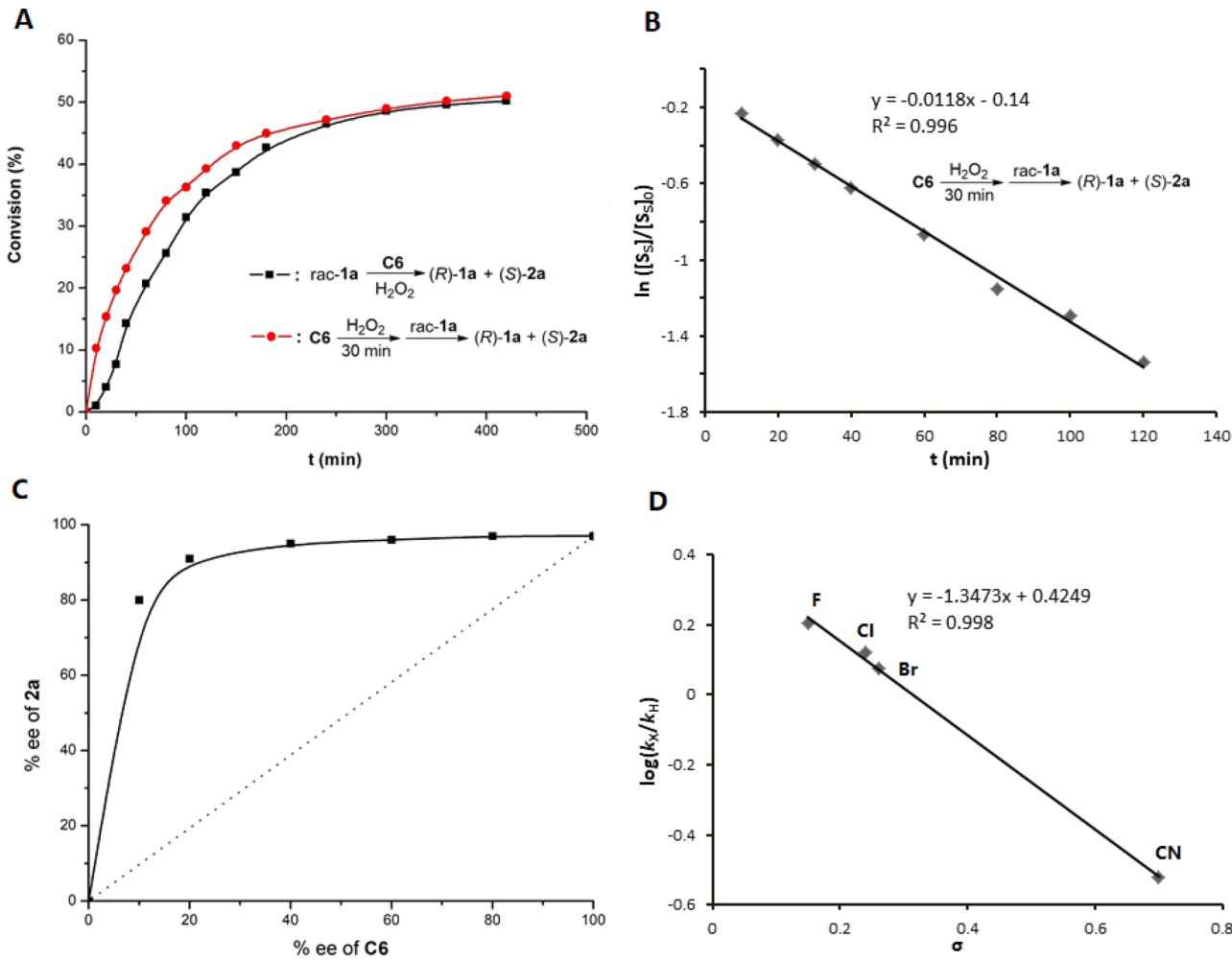

**Fig. 6 Mechanistic studies. A** Kinetic plots for the oxygenation of rac-**1a** with **C6**/$H_2O_2$. **B** ln([$S_S$]/[$S_S$]$_0$) versus time dependences for the oxygenation of (S)-**1a**. [$S_S$]$_0$ = [(S)-**1a**]$_0$ = 0.05 M, [$S_S$] = [(S)-**1a**] ≈ [0.5 – C][rac-**1a**]$_0$, [rac-**1a**]$_0$ = 0.1 M, conversion = C = (ee$_{1a}$)/(ee$_{1a}$ + ee$_{2a}$). **C** Plot of the enantiomeric excess of **2a** versus the enantiomeric excess of **C6** at 50% conversion. The dotted line symbolizes the linear correlation. **D** Hammett plot of log($k_X$/$k_H$) vs $\sigma$ for the competitive oxidation of $C_5$-substituted indolines by **C6**/$H_2O_2$.

participated in the asymmetric oxygenation reaction, furnishing (2R, 3R)-**9** in 45% yield with 96% ee together with (2S, 3S)-**10** in 47% yield with 91% ee (s = 85) (Fig. 5d). $C_2$-Cyclopropane-substituted indoline **11** was discovered as a general pharmacophore for gonadotropin-releasing hormone (GnRH) antagonists (Fig. 5e)[53]. Asymmetric oxidation of rac-**12** proved to be relatively sluggish, and when 2.0 equiv of aqueous $H_2O_2$ was used, (R)-**12** was recovered in 49% yield with 83% ee (s = 23) together with over-oxidized nitrone **13** in 45% yield (Fig. 5e).

## Discussion

To get a preliminary understanding of the catalysis role of titanium(salalen) **C6** in asymmetric oxygenation of indolines, conversion (%) of rac-**1a** was plotted against time (min) for reaction under standard conditions and that with pre-mixed catalyst and $H_2O_2$, respectively (Fig. 6A). For the standard reaction, conversion followed a sigmoidal curve, and an induction period of about 30 min was observed. Such induction period was not observed in the reaction using pre-mixed **C6** and $H_2O_2$, suggesting that the generation of the real active species for oxygenation reaction is a slow process. Moreover, the latter reaction exhibited an apparent steady-state regime, resulting in linear ln([$S_S$]/[$S_S$]$_0$) vs. time dependence (Fig. 6B).

Control experiments were performed to further understand the identity of the generated species by mixing **C6** with $H_2O_2$. The relationship between the excesses of **2a** and the catalyst **C6** was investigated, and a positive nonlinear relationship was observed, suggesting the possible intervention of a dimeric complex (Fig. 6C)[54,55]. ESI mass-spectrometry analysis of the mixture of di-μ-oxo titanium(salalen) **C6** and 30% aqueous $H_2O_2$ after 30 min showed the peak of 1797.6, which is equal to [$M_{C6}$ + O + H]$^+$, implying the formation of a μ-oxo-μ-peroxo species **S** (Fig. 7a)[56]. However, a lower intensity of this peak was observed for mixing **C6** and $H_2O_2$ after 15 min, implying that the formation of **S** might be a slow process. No reaction was observed when mixing one equiv of **S** with rac-**1a** (Fig. 7b). However, the addition of aqueous hydrogen peroxide rendered **S** catalytically active with comparable results to **C6** (Fig. 7c). These results clearly indicate that **S** is neither the real active species nor a dead-end species but an active intermediate. Crossover experiments involving two different catalysts **C4** and **C6** were next performed. Mass spectra analysis of equivalent **C4** and **C6** in the presence of 30% aqueous $H_2O_2$ showed the peak of 1445.5, which is equal to [$M_{C4}$/2 + $M_{C6}$/2 + O + H]$^+$, suggesting the generation of a crossover dimerization peroxo complex (Fig. 7d). No such peak was detected during the mass spectra analysis of the mixture of **C4** and **C6** in the absence of $H_2O_2$ (Fig. 7e). These observations indicated that a disassembly

**a**) ESI-MS analysis of the mixture of **C6** and H$_2$O$_2$

**C6**
[M + H]$^+$ = 1781.6

$\xrightarrow[\text{CHCl}_3]{\text{H}_2\text{O}_2}$

**S**
μ-oxo-μ-peroxo species
*m/z* = **1797.6**

**b**) The oxidation reactivity of stoichiometric **S** without H$_2$O$_2$

rac-**1a** $\xrightarrow[\text{CHCl}_3]{\textbf{S} \ (1.0 \ \text{eq})}$ no reaction

**c**) The oxidation catalysis reactivity of **S** with H$_2$O$_2$

rac-**1a** $\xrightarrow[\text{CHCl}_3]{\substack{\textbf{S} \ (1 \ \text{mol} \%) \\ \text{H}_2\text{O}_2 \ (1.0 \ \text{eq})}}$ (*R*)-**1a**  +  (*S*)-**2a**
44%, 91% ee    46%, 90% ee
**s = 60**

**d**) ESI-MS analysis of the mixture of **C4**, **C6**, and H$_2$O$_2$

**C4**
[M + H]$^+$ = 1077.3

+ **C6**   $\xrightarrow[\text{CHCl}_3]{\text{H}_2\text{O}_2}$   crossover dimerization
peroxo species
*m/z* = **1445.5**

**e**) ESI-MS analysis of the mixture of **C4** and **C6** without H$_2$O$_2$

**C4**  +  **C6**  $\xrightarrow[\text{CHCl}_3]{\text{H}_2\text{O}}$  no reaction

**Fig. 7 Control experiments of titanium(salalen) catalyst with aqueous H$_2$O$_2$. a** ESI-MS analysis of the mixture of **C6** and H$_2$O$_2$. **b** The oxidation reactivity of stoichiometric **S** without H$_2$O$_2$. **c** The oxidation catalysis reactivity of **S** with H$_2$O$_2$. **d** ESI-MS analysis of the mixture of **C4**, **C6**, and H$_2$O$_2$. **e** ESI-MS analysis of the mixture of **C4** and **C6** without H$_2$O$_2$.

and reassembly process might occur after the formation of μ-oxo-μ-peroxo species **S**. We envisioned that the real active species involved in the catalytic cycle might be the disassembled monomeric peroxo Ti(salalen) complex[57].

Asymmetric oxygenation of racemic indolines with various kinds of C$_5$-substituents (X) on indoline arene were performed and the reaction rates were dependent on the electronic effects of the substituents (see the Supplementary Information for details)[58,59]. The Hammett plot ($\log(k_X/k_H)$ *versus* $\sigma$) for the competitive oxidation of **1a** and respective variants was exhibited in Fig. 6D. The observed plot displayed linear correlation with a $\rho$ value of $-1.347$ ($R^2 = 0.99$). Good linearity suggests that the oxidation proceeds through a single mechanism. The negative value of $\rho$ indicates a positive charge build-up on nitrogen in the transition state[60]. The use of Hammett parameter $\sigma^+$ gave a

relatively poorer correlation ($R^2 = 0.97$, see the Supplementary Information for details). These data are consistent with a concerted mechanism, in which an electrophilic active oxygen species might be directly attacked by a nucleophilic nitrogen[61].

Based on the above studies and literature survey, a plausible mechanistic pathway for asymmetric oxygenation of racemic indolines was suggested (Fig. 8). Treating di-μ-oxo titanium(salalen) **C6** with hydrogen peroxide gave μ-oxo-μ-peroxo species **S**, which is the incubation period for the oxygenation process. In the presence of hydrogen peroxide, relatively stable **S** underwent a disassembly process providing monomeric peroxo Ti(salalen) complex **14**, which was proposed to be the real active species involved in the catalytic cycle[62,63]. The asymmetric nucleophilic attack of the nitrogen of indoline rac-**1a** onto electrophilic oxygen of chiral complex **14** through **15** generated complex **16**. **16**

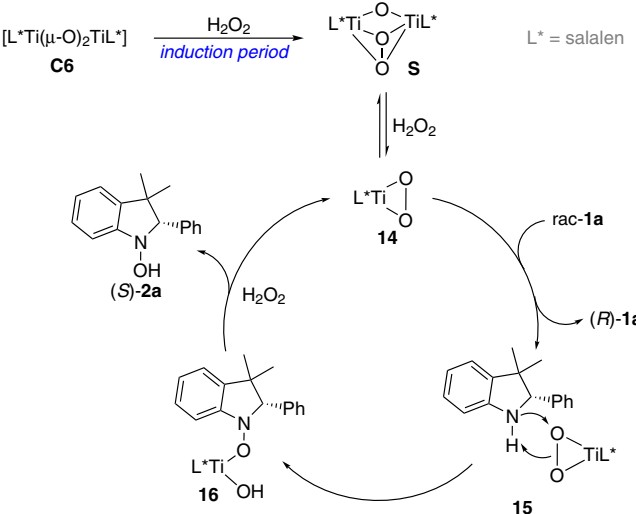

**Fig. 8 Proposed mechanistic pathway.** The possible reaction pathway based on our studies and the previous literatures.

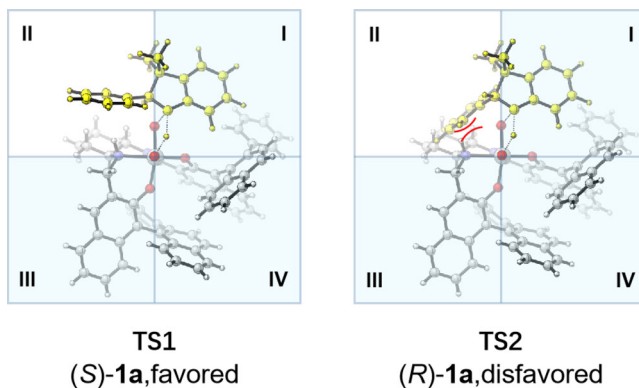

**Fig. 9 Proposed transition state models.** The transition state **TS1** for (S)-**1a** is favored. The transition state **TS2** for (R)-**1a** is disfavored.

reacted with hydrogen peroxide providing hydroxylamine **2a** together with **14** for the catalytic cycle. According to the absolute configuration of recovered indolines, we envisioned that amine (S)-**1a** was oxidized more preferentially than (R)-**1a**, and therefore unreacted (R)-**1a** was isolated with high enantioselectivity.

The stereochemical induction model was suggested in Fig. 9. The structure of monomeric peroxo Ti(salalen) complex **14** is proposed based on the crystal structure of di-μ-oxo titanium (salalen) **C6**[64,63]. The quadrant diagrams elaborate the origins of the chiral recognition. The steric repulsion between two (R)-binaphthyl unites and indoline **1a** prevents the nucleophilic attack of the nitrogen from the third and fourth quadrants. In addition, locating the less sterically demanding phenyl moiety of indoline skeleton in the first quadrant is preferred to avoid the steric repulsion between the (R)-binaphthyl unite and the α-substituent of **1a**. The transition state **TS2** for (R)-**1a** has obvious steric repulsion between the α-phenyl substituent of indoline with the cyclohexane moiety of salalen ligand. Such steric repulsion is absent in the transition state **TS1** for (S)-**1a**. Therefore, (S)-**1a** was oxidized more preferentially than (R)-**1a**, and unreacted (R)-**1a** was recovered with high enantioselectivity.

In summary, an oxidative KR of secondary amines based on N–O bond formation is reported. The practical titanium(salalen) catalyzed asymmetric oxygenation with environmentally benign

hydrogen peroxide as oxidant is applicable to a range of indolines with multiple stereocenters and diverse substituent patterns in high efficiency with efficient chemoselectivity and enantio-discrimination. Late-stage asymmetric oxygenation of bioactive molecules that are otherwise difficult to synthesize was further explored. The KR of secondary amines based on N–O bond formation described herein represents an advance in the field of asymmetric oxidation.

## Methods
**General procedure.** To a solution of racemic indoline (0.1 mmol, 1.0 eq) in CHCl$_3$ (1.0 mL) was added 30% aqueous hydrogen peroxide (0.1 mmol, 10 μL, 1.0 eq) and **C6** (0.001 mmol, 1.8 mg, 1 mmol %) at room temperature. The reaction was vigorously stirred for 4–12 h. Then the mixture was diluted with CH$_2$Cl$_2$ (20 mL), washed with water (10 mL), dried over MgSO$_4$, filtered, and concentrated. The residue was purified by silica gel chromatography (EtOAc/petroleum ether) to give the desired product.

## Data availability
The authors declare that the data supporting the findings of this study are available within the article and its Supplementary Information files. Extra data are available from the corresponding author upon request.

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

## Acknowledgements

Financial supports were provided by the National Science Foundation of China (21971148, 21722204), Shenzhen Special Funds (JCYJ20190807093805572), and Youth Interdiscipline Innovative Research Group of Shandong University (2020QNQT009).

## Author contributions

G.W. conducted the asymmetric oxygenation experiments and mechanistic studies; R.L. and C.H. prepared the substrates; L.L. designed the experiments and wrote the paper.

## Competing interests

The authors declare no competing interests.
