## [Peer Review File · Nature Communications]

REVIEWER COMMENTS

Reviewer #1 (Remarks to the Author):

I am very excited to read the full text. Indeed, catalytic kinetic resolution of amines represents a longstanding challenge in chemical synthesis. The authors described the kinetic resolution of secondary amines through oxygenation to produce enantiopure hydroxylamines involving N–O bond formation firstly.

In particular, they used environmentally benign hydrogen peroxide as oxidant and a range of racemic indolines with multiple stereocenters and diverse substituent patterns were resolved in high efficiency with extremely efficient chemoselectivity and enantio-discrimination. I think the results based on their detailed and comprehensive research can be published in Nature Communications. However, it is hoped that the author will explain the stability of the catalyst and discuss the structure-activity relationship of the catalyst more deeply. Whether can try to explain through the theory calculation.

Reviewer #2 (Remarks to the Author):

This paper dealt with the kinetic resolution of indolines via oxidation of secondary amines to hydroxylamine by using Ti-salen derivatives as catalyst and hydrogen peroxide as oxidant. The reaction is facile and efficient with high S-factor, affording both hydroxylamine and starting material having one or two chiral centers in high yields with high ee. The authors did good mechanism investigations by kinetics and controlled experiments and rational mechanism was proposed. Chiral amines are important in organic synthesis, biologically active molecule studies and pharmaceuticals. This work provides an easy way to chiral amines, represents an important advancement in chiral amine synthesis. On the other hand, kinetic resolution is an useful means in asymmetric synthesis though the yield is less than 50% because either of pair of enantiomers can be obtained by kinetic resolution. The present study realized the kinetic resolution of amine via oxidation in high efficiency, which should be an excellent example in kinetic resolution. With above consideration, the publication is recommended after some revision.

1. The authors should discuss the stereochemistry and provide the possible transition state of the reaction.
2. Some substrates having different kind of functional groups may be tested to show the tolerance of the reaction further.
3. The authors may mention the shortages of kinetic resolution in addition to its advantages in introduction part.

Reviewer #3 (Remarks to the Author):

Indoline is an important scaffold in the fields of organic chemistry and medicinal chemistry. A number of methods have been already reported for the enantioselective synthesis of indolines. Kinetic resolution of secondary amines is mostly achieved by the N-acylation strategy. Lei Liu and co-workers describe in this manuscript kinetic resolution of indolines by the asymmetric oxygen atom transfer (OAT) reaction by use of chiral titanium(salen) complex bearing two (R)-binaphthyl units. High efficiency in the kinetic resolution was attained and a range of indolines bearing one stereocenter and two stereocenters were obtained in excellent enantioselectivities. Mechanistic study elucidated the mechanism. One of the limitations of this protocol is that 3,3-disubstitution is critical.

This reviewer recommends publication of the manuscript in Nat. Commun. after addressing following issues.

(1) Compound 7j in Scheme 4 appears to be unstable due to the presence of benzylic C3-H bond. It is curious which kind of reaction took place under the conditions.

(2) Following paper of Toste for the deracemization of indoline should be included.

Lackner, A. D.; Samant, A. V.; Toste, F. D. Single-Operation Deracemization of 3H-Indolines and Tetrahydroquinolines Enabled by Phase Separation. *J. Am. Chem. Soc.* 2013, 135, 14090–14093.

(3) There are several mistakes, grammatical errors, and typos.

Table 1. Catalysts C8, C9, are C10 are missing, should read C7?

Page 1, line 7 from the bottom, templet should read template?

Page 2, line 8 and 6 from the bottom, please catalysis to catalytic.

Page 5, line 5, please change germinal to geminal.

Thank you for your careful review of the manuscript entitled “Kinetic Resolution of Indolines by Asymmetric Hydroxylamine Formation” (NCOMMS-20-48155), and kind invitation to submit a revised manuscript. We have modified the manuscript in response to the extensive and insightful reviewer comments, and hope the revised manuscript to be considered for publication as an article in the Nature Communications.

As instructed, we have attempted to succinctly explain changes made in reaction to all comments. Here we reply to each comment in point-by-point fashion.

Reviewer 1 comment:

Comment:

It is hoped that the author will explain the stability of the catalyst and discuss the structure-activity relationship of the catalyst more deeply. Whether can try to explain through the theory calculation.

Response:

Thanks for the reviewer’s comment. The di- μ -oxo titanium(salalen) catalyst is pretty stable, and can be stored at ambient temperature for several months. Such description has been included in the revised Supplementary Information.

The monomeric peroxo Ti(salalen) complex **14** in Fig. 8 was proposed as a real active species involved in the catalytic cycle based on our control experiments and literature survey. However, such species has not been verified by literatures. Therefore, we think that the calculations based on such active species would not provide sufficient support to elucidate the structure-activity relationship of the catalyst.

Scheme R1. Proposed transition state models.

According to the reviewer’s comment, a stereochemical induction was suggested in Scheme R1 (Fig. 9 in revised manuscript). The structure of reactive monomeric peroxo Ti(salalen) complex is proposed based on the crystal structure of di- μ -oxo titanium(salalen) **C6**. The quadrant diagrams elaborate the origins of the chiral

recognition. The steric repulsion between two (*R*)-binaphthyl unites and indoline **1a** prevents the nucleophilic attack of the nitrogen from the third and fourth quadrants. In addition, locating the less sterically demanding phenyl moiety of indoline skeleton in the first quadrant is preferred to avoid the steric repulsion between the (*R*)-binaphthyl unite and the α -substituent of **1a**. The transition state **TS2** for (*R*)-**1a** has obvious steric repulsion between the α -phenyl substituent of indoline with the cyclohexane moiety of salalen ligand. Such steric repulsion is absent in the transition state **TS1** for (*S*)-**1a**. Therefore, (*S*)-**1a** was oxidized more preferentially than (*R*)-**1a**, and unreacted (*R*)-**1a** was recovered with high enantioselectivity.

Reviewer 2 comment:

Comment-1:

The authors should discuss the stereochemistry and provide the possible transition state of the reaction.

Response-1:

Scheme R1. Proposed transition state models.

We appreciated for the reviewer's comment. A stereochemical induction was suggested in Scheme R1 (Fig. 9 in revised manuscript). The structure of reactive monomeric peroxy Ti(salalen) complex is proposed based on the crystal structure of di- μ -oxo titanium(salalen) **C6**. The quadrant diagrams elaborate the origins of the chiral recognition. The steric repulsion between two (*R*)-binaphthyl unites and indoline **1a** prevents the nucleophilic attack of the nitrogen from the third and fourth quadrants. In addition, locating the less sterically demanding phenyl moiety of indoline skeleton in the first quadrant is preferred to avoid the steric repulsion between the (*R*)-binaphthyl unite and the α -substituent of **1a**. The transition state **TS2** for (*R*)-**1a** has obvious steric repulsion between the α -phenyl substituent of indoline with the cyclohexane moiety of salalen ligand. Such steric repulsion is absent in the transition state **TS1** for (*S*)-**1a**. Therefore, (*S*)-**1a** was oxidized more preferentially than (*R*)-**1a**, and unreacted (*R*)-**1a** was recovered with high enantioselectivity.

Comment-2:

Some substrates having different kind of functional groups may be tested to show the tolerance of the reaction further.

Response-2:

As suggested by the reviewer, two substrates bearing respective carboxylic acid ester (**1r**) and terminal alkyne (**1s**) were prepared (Fig. 2). Both substrates exhibit high level of chiral recognition, with respective selectivity factor of 50 and 70, thus further demonstrating the good function group tolerance of the KR strategy.

Comment-3:

The authors may mention the shortages of kinetic resolution in addition to its advantages in introduction part.

Response-3:

Thanks for the reviewer's comment. The shortages of kinetic resolution have been included in the introduction part as "though the theoretical yield of the expected optically pure target can never exceed a limit of 50%."

Reviewer 3 comment:**Comment-1:**

Compound **7j** in Fig. 4 appears to be unstable due to the presence of benzylic C3-H bond. It is curious which kind of reaction took place under the conditions.

Response:

Thanks for the reviewer's suggestion. As encouraged by the reviewer, the reaction was re-performed and analyzed carefully. Under the oxidation conditions, the oxidized hydroxylamine **7j** was further oxidized to **7ja** and **7jb** (Scheme R2). **7jb** was stable and fully characterized. However, during the characterization of **7ja**, air

**Scheme R2.** Oxidation studies of hydroxylamine **7j**

oxidation of **7ja** to **7jb** was observed, which is in accord with the literature report (ARKIVOC, Gainesville, FL, United States, **2003**, 8, 102-111). These observations suggested that the geminal disubstitution at C3-position of indolines should be crucial to obtaining the stable hydroxylamine by avoiding the further oxidation processes. We would thank the reviewer again to encourage us to further understand the reaction details.

Comment:

Following paper of Toste for the deracemization of indoline should be included. Lackner, A. D.; Samant, A. V.; Toste, F. D. Single-Operation Deracemization of

3H-Indolines and Tetrahydroquinolines Enabled by Phase Separation. J. Am. Chem. Soc. 2013, 135, 14090–14093.

Response:

As suggested by the reviewer, redox deracemization of amines including the seminal work from the Trost group together with the other two literatures from respective Zhou and our groups have been cited in the revised manuscript.

Comment:

There are several mistakes, grammatical errors, and typos.

Table 1. Catalysts C8, C9, are C10 are missing, should read C7?

Page 1, line 7 from the bottom, templet should read template?

Page 2, line 8 and 6 from the bottom, please catalysis to catalytic.

Page 5, line 5, please change germinal to geminal.

Response:

We appreciate for the reviewer's careful proofreading. As suggested by the reviewer, all the above mistakes and grammatical errors have been corrected in the revised manuscript.

We thank you for your constructive comments and feedback. We tried our best to address each of your points in detail. We feel the quality of the revised manuscript is much improved and hope you agree.

Thank you for your efforts on behalf of this manuscript.

With sincere regards,

Lei Liu,
Shandong University

REVIEWERS' COMMENTS

Reviewer #2 (Remarks to the Author):

This revised revised manuscript addressed well all issues I raised and thus the publication is recommended.

Reviewer #3 (Remarks to the Author):

The revised manuscript by Liu and co-workers has been improved and this reviewer recommends publication of the manuscript in the present form.